# Filamentous fungal associates of the alder bark beetle, *Alniphagus aspericollis*, including an undescribed species of *Neonectria*

**Gervais Y. S. Lee**¤, **Debra L. Wertman**, **Allan L. Carroll***, **Richard C. Hamelin**

Department of Forest and Conservation Sciences, The University of British Columbia, Vancouver, British Columbia, Canada

¤ Current address: Singapore Botanic Gardens, National Parks Board, Singapore, Singapore
* allan.carroll@ubc.ca

## Abstract

Bark beetles (Coleoptera: Curculionidae; Scolytinae) are tree-infesting insects that consume subcortical tissues and fungi. Species capable of killing their host trees are most commonly associated with conifers, as very few bark beetle species infest and kill hardwood hosts directly. The alder bark beetle, *Alniphagus aspericollis*, is a hardwood-killing bark beetle that colonizes and kills red alder, *Alnus rubra*. Conifer-killing bark beetles have well-known associations with symbiotic ophiostomatoid fungi that facilitate their life histories, but it is unknown whether *A. aspericollis* has any fungal associates. This study was conducted to identify any consistent filamentous fungal associates of *A. aspericollis* and characterize the consistency of observed beetle–fungus relationships. Beetles and gallery phloem samples were collected from seven sites throughout the Greater Vancouver region in British Columbia, Canada. Filamentous fungi were isolated from these samples and identified by DNA barcoding using the internal transcribed spacer (ITS) region and other barcode regions for resolution to the species-level for the most dominant isolates. The most common fungal associate was a previously undescribed *Neonectria major*-like fungus, *Neonectria* sp. nov., which was isolated from ~67% of adult beetles, ~59% of phloem samples, and ~94% of the beetle-infested trees. *Ophiostoma quercus* was isolated from ~28% of adult beetles, ~9% of phloem samples, and ~56% of infested trees and deemed a casual associate of *A. aspericollis*, while a putatively novel species of *Ophiostoma* was more infrequently isolated from *A. aspericollis* and its galleries. *Cadophora spadicis*, a new record for red alder, was rarely isolated and is probably coincidentally carried by *A. aspericollis*. Overall, *A. aspericollis* was only loosely associated with ophiostomatoid fungi, suggesting that these fungi have little ecological significance in the beetle–tree interaction, while *Neonectria* sp. nov. may be a symbiote of *A. aspericollis* that is vectored by the beetle.

**Data Availability Statement:** Nucleotide sequence data have been deposited into the GenBank database under the accession numbers OP739206-OP739243 and OP787825-OP787865.

**Funding:** ALC: RGPIN-2015-04376, Natural Sciences and Engineering Research Council of Canada, Discovery Grant, https://www.nserc-crsng.gc.ca/ RCH: RGPIN-2015-05279, Natural Sciences and Engineering Research Council of Canada, Discovery Grant, https://www.nserc-crsng.gc.ca/ The funding agency did not play any role in the study design, data collection and analysis, decision to publish, or preparation of the manuscript.

**Competing interests:** The authors have declared that no competing interests exist.

## Introduction

Bark beetles of the subfamily Scolytinae (Coleoptera: Curculionidae) are subcortical herbivore-fungivores that occur in forested ecosystems worldwide. They range from monophagous to oligophagous, breeding in the phloem-cambium tissues of their tree hosts where their larvae feed and pupate [1, 2]. Most bark beetles are secondary invaders and only target stressed or moribund trees that have been weakened by other biotic or abiotic stressors, such as defoliators and drought [3–5]. Some bark beetle species, however, occasionally build epidemic populations that overwhelm healthy, live trees via pheromone-mediated mass attacks [4, 6]. These irruptive bark beetles have historically caused widespread tree-mortality events at the landscape scale, resulting in significant economic and ecological consequences [7–11].

Bark beetles evolved alongside their symbiotic fungal partners [12]. These fungi may either be carried phoretically on the exoskeleton of the beetles [13–15] or contained within specialised structures called mycangia that occur within the mouthparts, thorax, or elytra of the beetles [16–18]. Bark beetle–fungus relationships have typically been considered as mutually beneficial to both the beetle and fungus partners [19]. Beetles facilitate the transmission of fungi from one host tree to another, thereby enabling them to colonise fresh resources ahead of other late successional saprotrophs [20]. In return, beetles benefit from improved nutrition to developing larvae within fungal-colonized phloem [21–23], and/or phytopathogenic activity by fungi that exhausts host tree defenses and blocks water conduction in the xylem [5, 24–26]. Reviews by Paine et al. [5] and Six and Wingfield [20] have highlighted that not all bark beetle–fungus relationships are necessarily mutualistic. Rather, these symbioses comprise a spectrum ranging from true mutualisms, to commensalism, or even antagonistic associations that can be obligatory, facultative, or purely coincidental [5, 20, 27]. The nature of bark beetle–fungus relationships apparently vary relative to host tree vigor, the specific fungal associate, and the life history of the beetle [5, 28]. The most well-studied bark beetle-symbiotic fungi are found in conifer systems and belong to the ascomycete order Ophiostomatales, which includes the genera *Ophiostoma*, *Grosmannia*, *Leptographium*, and *Ceratocystiopsis* [20, 29, 30]. The genera *Ceratocystis* and *Bretziella* (Order: Microascales) also contains "ophiostomatoid" fungi associated with bark beetles.

The majority of bark beetle species capable of infesting and killing their host trees across landscapes are associated with conifers [31]. Most bark beetles that feed on hardwoods are not stand-level tree killers and instead colonise moribund hosts or dying tree limbs. Even in the well-known case of Dutch elm disease, *Ophiostoma ulmi* and *O. novo-ulmi*, vectored by *Scolytus* spp. bark beetles, and likely also in oak wilt, *Bretziella fagacearum*, which is associated with *Pseudopityophthorus* spp. bark beetles, healthy host trees are first infected with the pathogens during adult beetle maturation feeding (e.g., in branch crotches) and subcortical invasion by the beetles does not occur until weeks after initial inoculation, when the host tree is already dying from the disease [32–34]. Ohmart [31] hypothesized that the evolution of hardwood-killing bark beetles might have been hampered, either because induced defenses of hardwoods were too complex, or because the nutritional benefits gained by hardwood-invading bark beetles were outweighed by the high physiological costs of colonization. Given that the fungal symbionts of bark beetles may play a role in overcoming host tree defenses, examining the fungal communities of the few apparent hardwood-killing beetles could provide insight into how these beetles kill trees.

The alder bark beetle, *Alniphagus aspericollis* (LeConte) attacks red alder, *Alnus rubra* (Bong.), an early successional nitrogen-fixing hardwood found throughout the coastal Pacific Northwest [35–38]. Though *A. aspericollis* is frequently found in alder slash, it is able to kill live red alders that are apparently healthy or previously stressed by other factors [35, 36].

Symbiotic fungal partners of *A. aspericollis*, if any, are yet to be discovered. Borden [36] observed that infestation by *A. aspericollis* could precede wood rot but did not suggest that fungal inoculation was directly or indirectly facilitated by *A. aspericollis*. Previous studies of the fungal communities of red alder in the Pacific Northwest have recorded various species of saprotrophic decay fungi and non-ophiostomatoid pathogens but no associations with *A. aspericollis* were reported [39, 40].

This study explores the filamentous fungal community associated with *A. aspericollis*. Our objectives were to determine if *A. aspericollis* is associated with fungi and characterize the consistency of any beetle-fungus relationships. Given the ability of *A. rubra* to fix nitrogen from the atmosphere via symbiotic root nodule actinomycetes [37, 38], we also tested the prediction that *A. aspericollis* would not require, and therefore not be consistently associated with, nitrogen-provisioning ophiostomatoid fungi in the same manner as conifer-feeding bark beetles that depend on these fungi to complete their development [21–23].

## Methods

### Beetle and gallery phloem collection

Adult *A. aspericollis* (Fig 1) were collected from seven sites across the Greater Vancouver region in British Columbia, Canada between 14 July and 26 August 2015 (Fig 2). Research permits were obtained from Metro Vancouver Regional Parks (Fig 2 sites 2–4, 6), the District of West Vancouver (site 1), the City of Port Coquitlam (site 5), and the City of Burnaby (site 7). One to three infested red alder hosts were located at each site, and three adult beetles were obtained from under the bark of each tree (Figs 1 and 2). A total of 54 individual beetles were extracted from separate entrance holes and non-overlapping galleries. Larvae, pupae and teneral adults were also collected when present. All specimens were individually transferred into separate vials with ethanol-sterilized tweezers. A section of phloem (approximately 5 x 5 cm) was also extracted from each of the sampled beetle galleries (n = 54) using a hammer and chisel. All beetle and gallery phloem samples were transported back to the laboratory and refrigerated at 4°C until fungal isolations were conducted.

### Fungal isolations

Fungal isolations from the beetle and phloem samples were performed within one week of collection. Each beetle was thoroughly washed with 100 μl of sterile water and 50 μl of the resulting liquid inoculum was thinly spread onto malt extract agar (MEA) in a petri dish. From each phloem sample, five thinly-sliced chips (approximately 1 x 1 cm) were sectioned out with sterilized instruments, particularly around the entrance hole and gallery regions. Phloem chips were rapidly surface sterilized using 70% ethanol, rinsed thoroughly with sterile water, and plated onto water agar in a petri dish for incubation at room temperature.

Individual filamentous fungal colonies emerged in the initial isolation dishes within three days of incubation. Since the colonies in each dish were generally too numerous to be individually isolated, they were first sorted into different morphotypes based upon visible characteristics (e.g., colony diameter, texture, hyphal/spore colour, margin characteristics). Representative samples and variants of each morphotype were then sub-cultured onto MEA to ensure that maximum fungal diversity was captured per initial isolation dish. Sub-cultures were incubated at room temperature, and repeatedly sub-cultured until pure isolates were obtained. Numerous yeast and bacterial colonies were also observed in most of the initial isolation dishes but were not sub-cultured as these taxa were beyond the scope of this study.

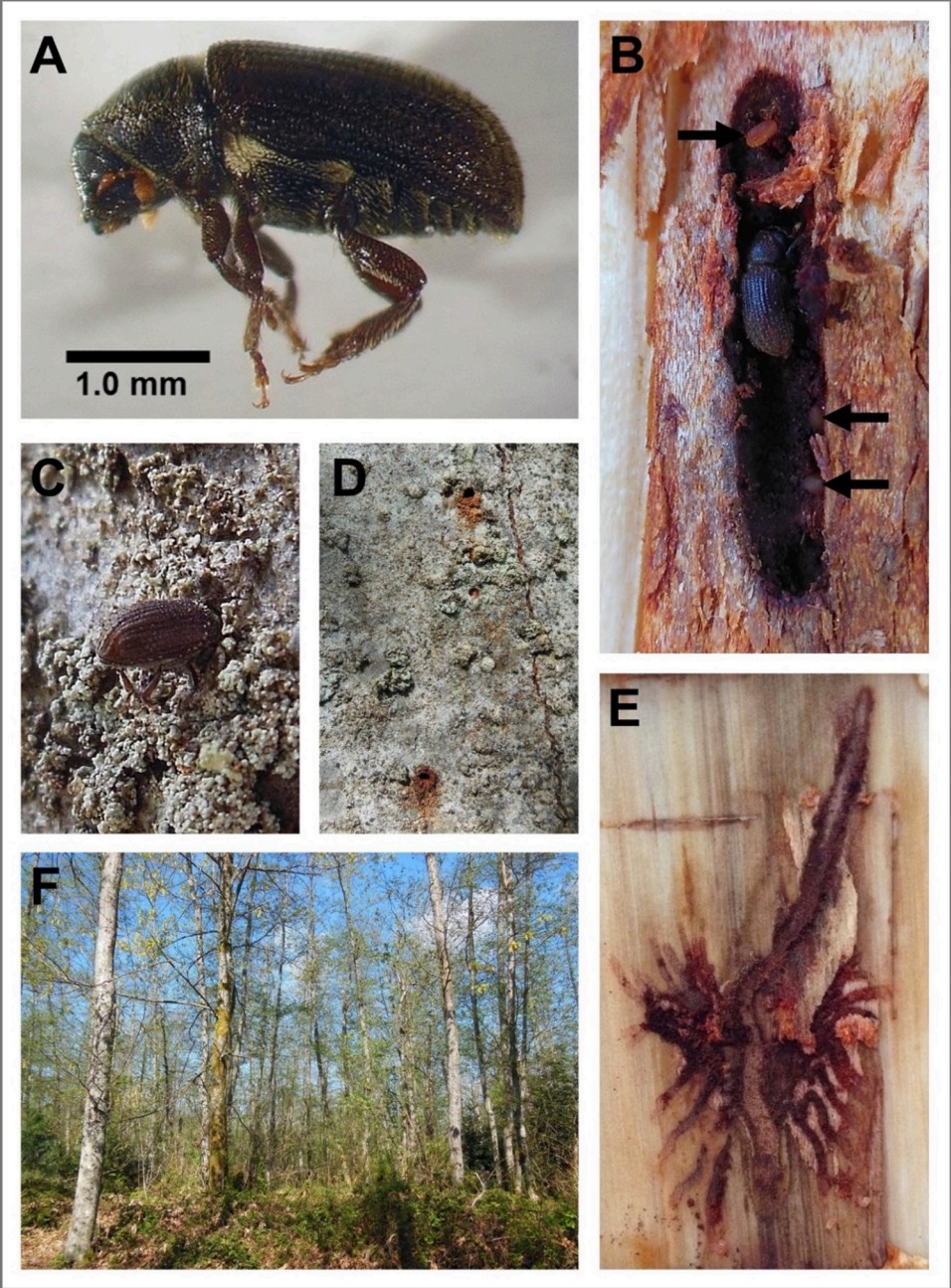

**Fig 1. The alder bark beetle, *Alniphagus aspericollis*, and its injuries on host red alder, *Alnus rubra*.** (A) Lateral view of an adult; (B) maternal gallery (eggs, including one dislodged from its niche at the top of the gallery, are indicated by arrows); (C) attack initiation; (D) newly-constructed gallery entrance holes in the bark of an infested tree; (E) maternal gallery and outward-radiating larval galleries in early stages of construction, with associated phloem staining; and (F) typical red alder stand where *A. aspericollis*-infested trees are located.

## DNA extraction, amplification, and sequencing

Mycelia from each pure isolate were frozen via liquid nitrogen and finely-ground. DNA samples were extracted with the Qiagen Dneasy® Plant Mini Kit (Qiagen Inc., Valencia, California, USA) according to the manufacturer's instructions. Internal transcribed spacer (ITS)

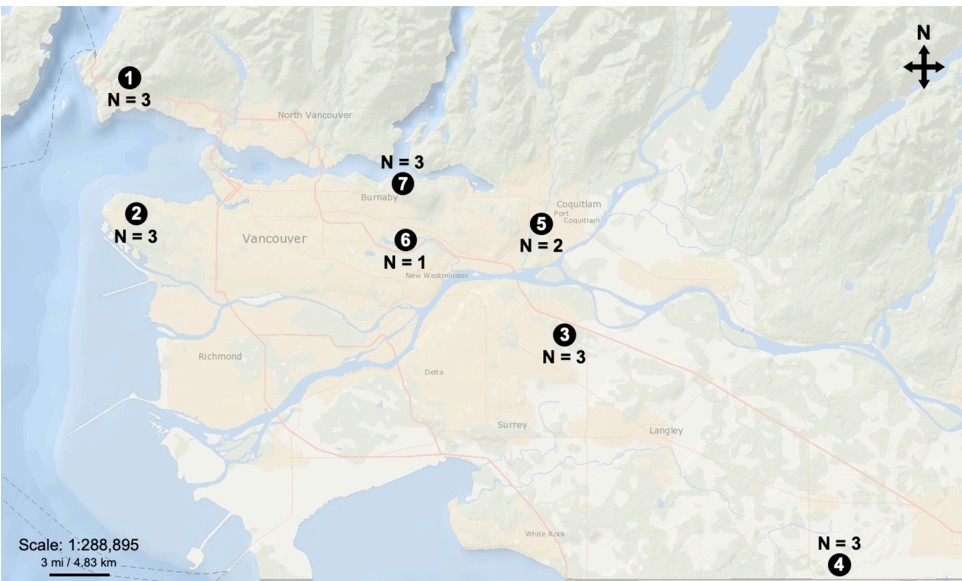

**Fig 2. Sampling locations within the Greater Vancouver region, British Columbia, Canada; (1) Cypress Mountain, (2) Pacific Spirit Regional Park, (3) Tynehead Regional Park, (4) Aldergrove Regional Park, (5) Gates Park, (6) Burnaby Lake Regional Park, and (7) Burnaby Mountain conservation area.** Map adapted from USGS National Map Viewer (2023) (Available from: http://viewer.nationalmap.gov/viewer/, accessed 7 March 2023). Three adult *Alniphagus aspericollis* and three corresponding gallery phloem samples were collected from each tree, with the number of trees sampled (N) per site indicated on the map. Specific locations and sampling dates for each site are as follows: (1) 49˚21'34" N, 123˚12'38" W, 14 July 2015; (2) 49˚16'16" N, 123˚14'20" W, 23 July 2015; (3) 49˚10'49" N, 122˚45'54" W, 22 July 2015; (4) 49˚0'56" N, 122˚27'58" W, 22 July 2015; (5) 49˚15'24" N, 122˚47'30" W, 24 August 2015; (6) 49˚14'46" N, 122˚55'50" W, 26 August 2015; (7) 49˚16'50" N, 122˚55'12" W, 26 August 2015.

rDNA gene region, which is a universal DNA barcode for all fungal species [41], was amplified from all samples using the primers ITS1F and ITS4 [42, 43]. PCR was performed with the ABI 2720 Thermal Cycler (Applied Biosystems, Foster City, California, USA) or the T100™ Thermal Cycler (Bio-Rad, Hercules, California, USA) using a 27 μl reaction system and the following thermal cycler program: 94˚C for 3 min, followed by 35 cycles of 94˚C for 30 s, 55˚C for 30 s, 72˚C for 1 min, and then held for 10 min at 72˚C. PCR amplicons were purified using Unifilter™ GF/C 384-well 100 μl microplates (Whatman Inc., Ann Arbor, Michigan, USA) and Sanger sequenced with the ABI 3730xl DNA Sequencer (Applied Biosystems, Foster City, California, USA) at the Centre de Recherche du CHUL (CHUQ), Université Laval (Quebec, Canada). ITS1F was used as the forward sequencing primer to produce single-strand sequences for all amplicons.

Additional gene regions were amplified and sequenced to confirm species identities for the fungal morphotypes most frequently isolated from beetles and phloem. For isolates from the genus *Neonectria*, two gene regions were targeted: i) the partial RNA polymerase II second largest subunit (RPB2) gene, amplified using nRPB2-147 and fRPB2-7cR primers [44, 45] and forward sequenced using nPRB2-147, and ii) the partial translation elongation factor 1-alpha (EF1-α) gene, amplified using EF1-728F and EF1-986RN primers [46, 47] and forward sequenced using EF1-728F. For isolates from the genus *Ophiostoma*, target regions included i) the partial beta-tubulin (BT) gene, amplified using Bt2A and Bt2B primers [48] and forward sequenced using Bt2A, and ii) the partial translation elongation factor 1-alpha (EF1-α) gene, amplified using EF1-728F and EF1-986R primers [46] and forward sequenced using EF1-728F. Targeted gene regions for isolates from the genus *Cadophora* were i) the partial translation elongation factor 1-alpha (EF1-α) gene, amplified using EF1-688F and EF1-1251R

primers [49] and forward sequenced using EF1-688F, and ii) the partial beta-tubulin (BT) gene, amplified using BTCadF and BTCadR [50] and forward sequenced using BTCadF.

## Sequence analysis and fungal identification

Raw sequences were edited and verified using BioEdit Sequence Alignment Editor software [51] version 7.1 (2011). Resulting ITS sequences were compared to existing GenBank sequence data using a BLAST similarity search (BLASTn) to identify each isolate to the genus- or species-level. To verify the species identities of the most frequently isolated morphotypes, single-locus phylogenetic trees were constructed via maximum likelihood analysis, using PhyML 3.0 [52] with the HKY85 substitution model. Phylogenies were constructed for each *Neonectria*, *Ophiostoma*, and *Cadophora* gene region. The species identity of each frequently-isolated morphotype was determined based on its phylogenetic placement relative to other closely-related species. Comparative phylogenetic data were derived from the most current relevant studies that also provided appropriate outgroups for tree rooting. For *Neonectria* isolates, *N. castaneicola* was chosen as the outgroup for the ITS dataset [45], while the more related *N. fuckeliana* was selected as the outgroup for the RPB2 dataset [47].

## Results

Fungi associated with *A. aspericollis* were identified to the genus or species-level for 89% of the fungal cultures obtained, based on DNA sequence homology of ≥ 99%. Comparative homology assessment of ITS barcode sequences identified fungi belonging to the genera *Neonectria*, *Ophiostoma*, *Graphilbum*, *Cadophora*, *Penicillium*, *Beauveria*, *Cladosporium*, *Acremonium*, *Cosmospora*, *Chondrostereum*, *Valsalnicola*, *Pezicula*, and *Umbelopsis* (Table 1). Lower DNA sequence homology matches (94–97%) were obtained for ITS sequences from fungal cultures of unknown Ascomycota (Table 2).

### Phylogenetic analyses and species identification

Isolates from the genus *Neonectria* (strains CGL 401 and CGL 402) were identified as an undescribed *N. major*-like species, hereafter *Neonectria* sp. nov., based upon morphological culture characteristics, existing host-specificity data, and by comparison of ITS, RPB2, and EF1-α sequences with those of closely-related *Neonectria* species (Fig 3) [45, 47, 56]. Overall, our phylogenetic analyses placed these *Neonectria* isolates most closely to *N. major* and *N. ditissima* but within a well-supported distinct clade (88–98% bootstrap support). DNA amplification of *Neonectria* sp. nov. isolates resulted in fragments of ~500 bp for the ITS barcode region, ~960 bp for the partial RPB2 gene region, and ~260 bp for the partial EF1-α gene region. The identity of *Neonectria* sp. nov. could not be fully resolved by phylogenetic analysis of the ITS barcode region, which grouped the isolates with species such as *N. major*, *N. ditissima*, and *N. neomacrospora* (Fig 3A). Phylogenetic analysis of the partial RPB2 gene region placed the *Neonectria* sp. nov. isolates within a single well-supported clade that was most closely related to *N. major* and *N. ditissima*, thereby distinguishing the species from *N. neomacrospora* and *N. ditissimopsis* (Fig 3B). Analysis of the partial EF1-α gene region also placed the isolates within a single well-supported clade that was related to, but distinct from, *N. ditissimopsis* and slightly more distantly related to *N. major* and *N. ditissima* (Fig 3C). *Neonectria major*, but not *N. ditissima*, has been isolated from *Alnus* hosts and the cultural characteristics of our *Neonectria* sp. nov. isolates (e.g., regularly-tufted aerial mycelia, saffron pigmentation on underside) (Fig 4A) are in agreement with existing descriptions of *N. major* [47], leading us to conclude that the *Neonectria* sp. nov. isolates observed during this study are *N. major*-like.

**Table 1. GenBank accession numbers for the ITS barcode sequences of filamentous fungi associated with *Alniphagus aspericollis*, corresponding closest BLAST (BLASTn) matches in GenBank and percent sequence homology.**

| Fungus group | Accession No(s). (subgroup ID) | Closest BLAST matches | Accession no. of closest matches | % similarity |
|---|---|---|---|---|
| ***Neonectria* fungi** | | | | |
| *Neonectria* sp. nov. | OP787854, OP787855 (NEO01) | *N. major*, type material | NR_121496 | 99 |
| | | "*N. galligena*"/*N. ditissima* a | JQ434582 | 99 |
| | | *N. major* | JF268767 | 99 |
| | | *N. ditissima* | HQ166494 | 99 |
| **Ophiostomatoid fungi** | | | | |
| *Ophiostoma quercus* | OP787843, OP787844, OP787845, OP787846, OP787847, OP787848, OP787852, OP787853 (OPQ01) | *O. quercus* | JX444663 | 100 |
| | | *O. quercus* | JQ319590 | 100 |
| | OP787849, OP787850, OP787851 (OPQ02) | *O. piceae* | KF531618 | 100 |
| | | *O. quercus* | KM100571 | 100 |
| | | *O. quercus* | FJ805463 | 100 |
| | OP787841, OP787842 (OPQ03) | *O. quercus* | JX444663 | 99 |
| | | *O. quercus* | JQ319590 | 99 |
| *Ophiostoma* sp. nov. | OP787840 (OPS1) | *O. catonianum* | EU443765 | 98 |
| | | *O. piceae* | KF531618 | 98 |
| | | *O. quercus* | KM100571 | 98 |
| *Graphilbum fragrans* | OP787856 (GRA01) | "*Pesotum fragrans*"/ *G. fragrans* b | JX444673 | 100 |
| | | *P. fragrans* | DQ396790 | 100 |
| | | "*Ophiostoma microcarpum*"/ *G. microcarpum* c | GU134170 | 100 |
| **Other fungi** | | | | |
| *Cadophora spadicis* | OP787860, OP787862 (CAD01) | *C.* sp. | HQ713754 | 100 |
| | | "*C. melinii*"/*C. spadicis* d | DQ404351 | 100 |
| | | *C. spadicis* | KM497029 | 100 |
| | | *C. malorum* | FJ903289 | 100 |
| | OP787861 (CAD02) | *C.* sp. | HQ713754 | 99 |
| | | "*C. melinii*"/*C. spadicis* d | DQ404351 | 99 |
| | | *C. spadicis* | KM497029 | 99 |
| | | *C. malorum* | FJ903289 | 99 |
| *Beauveria* spp. | OP787864 (BEA01) | *B. bassiana* | HQ880760 | 100 |
| | | *B.* sp. | GQ354221 | 100 |
| | OP787863 (BEA02) | *B. bassiana* | KP670430 | 99 |
| | | *B.* sp. | KF367502 | 99 |
| | | *B. pseudobassiana* | AB831659 | 99 |
| *Chondrostereum purpureum* | OP787859 (CHO01) | *C. purpureum* | KP195081 | 99 |
| | | *C. purpureum* | KT362916 | 99 |
| *Valsalnicola oxystoma* | OP787825 (VAL01) | *V. oxystoma* | JX519559 | 100 |

*(Continued)*

**Table 1.** (Continued)

| Fungus group | Accession No(s). (subgroup ID) | Closest BLAST matches | Accession no. of closest matches | % similarity |
|---|---|---|---|---|
| *Penicillium* spp. | OP787839 (PEN01) | *P. hoeksii* | KM189523 | 100 |
| | | *P. zhuangii* | KF769435 | 99 |
| | | *P. quercetorum* | KM189556 | 99 |
| | OP787838 (PEN02) | *P. pancosmium* | KP329841 | 100 |
| | | *P.* sp. | KF367512 | 100 |
| | | *P. waksmanii* | HQ607920 | 100 |
| | OP787837 (PEN03) | *P. waksmanii* | AY373940 | 100 |
| | | *P. ubiquetum* | JN617679 | 100 |
| | OP787836 (PEN04) | *P. nothofagi*, type material | NR_121518 | 100 |
| | | *P. godlewskii*, type material | NR_103658 | 100 |
| | | *P. cosmopolitanum* | JN617682 | 100 |
| | | *P. canescens* | AF034463 | 100 |
| | OP787835 (PEN05) | *P. westlingii* | JN617668 | 100 |
| | | *P. decaturense* | KT323158 | 100 |
| | OP787834 (PEN06) | *P. vancouverense*, type material | NR_121512 | 100 |
| | | *P. wellingtonense*, type material | NR_121519 | 99 |
| | | *P. pasqualense*, type material | NR_121513 | 99 |
| | OP787833 (PEN07) | *P. brevicompactum* | KP903609 | 100 |
| | | *P. bialowiezense* | KC427176 | 100 |
| | | *P. biourgeianum* | JX139727 | 100 |
| | OP787832 (PEN08) | *P.* sp. | KT121512 | 100 |
| | | *P. brevicompactum* | KP329602 | 100 |
| | OP787831 (PEN09) | *P. glabrum* | KR909187 | 100 |
| | | *P. spinulosum* | KF646101 | 100 |
| | | *P. adametzioides* | DQ681325 | 100 |
| | OP787830 (PEN10) | *P. kojigenum*, type material | NR_121253 | 100 |
| | | *P.* sp. | AM901674 | 100 |
| *Cladosporium* sp. | OP787858 (CLA01) | *C. cladosporioides* | KT824762 | 100 |
| | | *C.* sp. | KT315420 | 100 |
| | | *C. subuliforme* | KT600456 | 100 |
| | | *C. montecillanum* | KT600408 | 100 |
| *Acremonium* sp. | OP787865 (ACR01) | *A. furcatum* | JF311939 | 99 |
| | | Hypocreales sp. | HQ649873 | 99 |
| | | *A. furcatum*, type strain | AY378154 | 99 |
| *Cosmospora* sp. | OP787857 (COS01) | "*Acremonium* cf. *curvulum*" e | KM231818 | 100 |
| | | *C. viridescens* | JQ676175 | 100 |
| | | *C. viridescens* f | KJ676147 | 99 |
| *Pezicula* sp. | OP787829 (PEZ01) | *P. sporulosa* | KR859265 | 99 |
| | | *P. livida* | AF141180 | 99 |
| | | *P. californiae* | JX144758 | 99 |
| *Umbelopsis* sp. | OP787828 (UMB01) | *U. ramanniana* | KM017730 | 100 |
| | | *U. isabellina* | HQ630363 | 99 |

(*Continued*)

**Table 1.** (Continued)

| Fungus group | Accession No(s). (subgroup ID) | Closest BLAST matches | Accession no. of closest matches | % similarity |
|---|---|---|---|---|
| Unknown Ascomycota spp. | OP787827 (ASC01) | Roussollaceae sp. | KR014367 | 97 |
| | | Ascomycota sp. | KT004570 | 94 |
| | OP787826 (ASC02) | Ascomycete sp. | EU816396 | 96 |
| | | Ascomycota sp. | KF998994 | 95 |

a "*Neonectria galligena*" is now synonymous with *N. ditissima*, with the latter recognised as the official name [47].

b Renamed as *Graphilbum fragrans* [30].

c Renamed as *Graphilbum microcarpum* [30].

d Erroneously named as *C. melinii* by [53], corrected by [50] as *C. spadicis*.

e Placed into the genus *Cosmospora* by [54].

f From the phylogenetic study of *Cosmospora* by [55].

Isolates from the genus *Ophiostoma* were identified to the species-level based upon comparison of ITS, BT and EF1-α sequences with those of closely-related species [57–63] in the "*O. ulmi* complex" [64]. DNA amplification of *Ophiostoma* isolates resulted in fragments of ~600 bp for the ITS barcode region, ~400 bp for the partial BT gene region, and ~460–540 bp for the partial EF1-α gene region. Phylogenetic analyses conducted for all three gene regions split the *Ophiostoma* isolates into two distinct groups (Fig 5).

**Table 2.** Overall prevalence of filamentous fungus species associated with *Alniphagus aspericollis* across seven sampling sites throughout the Greater Vancouver region (British Columbia, Canada).

| Fungus | Adult beetles | | Gallery phloem | | Concurrent isolations | | Trees | |
|---|---|---|---|---|---|---|---|---|
| | n | % | n | % | n | % | n | % |
| ***Neonectria* fungi** | | | | | | | | |
| *Neonectria* sp. nov. | 36 | 66.7 | 32 | 59.3 | 29 | 53.7 | 17 | 94.4 |
| **Ophiostomatoid fungi** | | | | | | | | |
| *Ophiostoma quercus* | 15 | 27.7 | 5 | 9.3 | 3 | 5.6 | 10 | 55.6 |
| *Ophiostoma* sp. nov. | 3 | 5.6 | 2 | 3.7 | 0 | 0.0 | 3 | 16.7 |
| *Graphilbum fragrans* | 1 | 1.9 | 0 | 0.0 | 0 | 0.0 | 1 | 5.6 |
| **Other fungi** | | | | | | | | |
| *Cadophora spadicis* | 6 | 11.1 | 2 | 3.7 | 0 | 0.0 | 6 | 33.3 |
| *Beauveria* spp. | 3 | 5.6 | 0 | 0.0 | 0 | 0.0 | 2 | 11.1 |
| *Chondrostereum purpureum* | 0 | 0.0 | 1 | 1.9 | 0 | 0.0 | 1 | 5.6 |
| *Valsalnicola oxystoma* | 0 | 0.0 | 1 | 1.9 | 0 | 0.0 | 1 | 5.6 |
| *Penicillium* spp. | 12 | 22.2 | 4 | 7.4 | 1 | 1.9 | 7 | 38.9 |
| *Cladosporium* sp. | 2 | 3.7 | 0 | 0.0 | 0 | 0.0 | 2 | 11.1 |
| *Acremonium* sp. | 1 | 1.9 | 0 | 0.0 | 0 | 0.0 | 1 | 5.6 |
| *Cosmospora* sp. | 1 | 1.9 | 0 | 0.0 | 0 | 0.0 | 1 | 5.6 |
| *Pezicula* sp. | 0 | 0.0 | 1 | 1.9 | 0 | 0.0 | 1 | 5.6 |
| *Umbelopsis* sp. | 0 | 0.0 | 1 | 1.9 | 0 | 0.0 | 1 | 5.6 |
| Unknown Ascomycota spp. | 2 | 3.7 | 2 | 3.7 | 0 | 0.0 | 3 | 16.7 |

Frequency of isolation data are shown for adult beetles (n = 54), gallery phloem samples (n = 54), galleries (n = 54) with concurrent isolations from both adult beetles and phloem samples, and beetle-infested trees (n = 18) with isolations from either adult beetles or phloem samples.

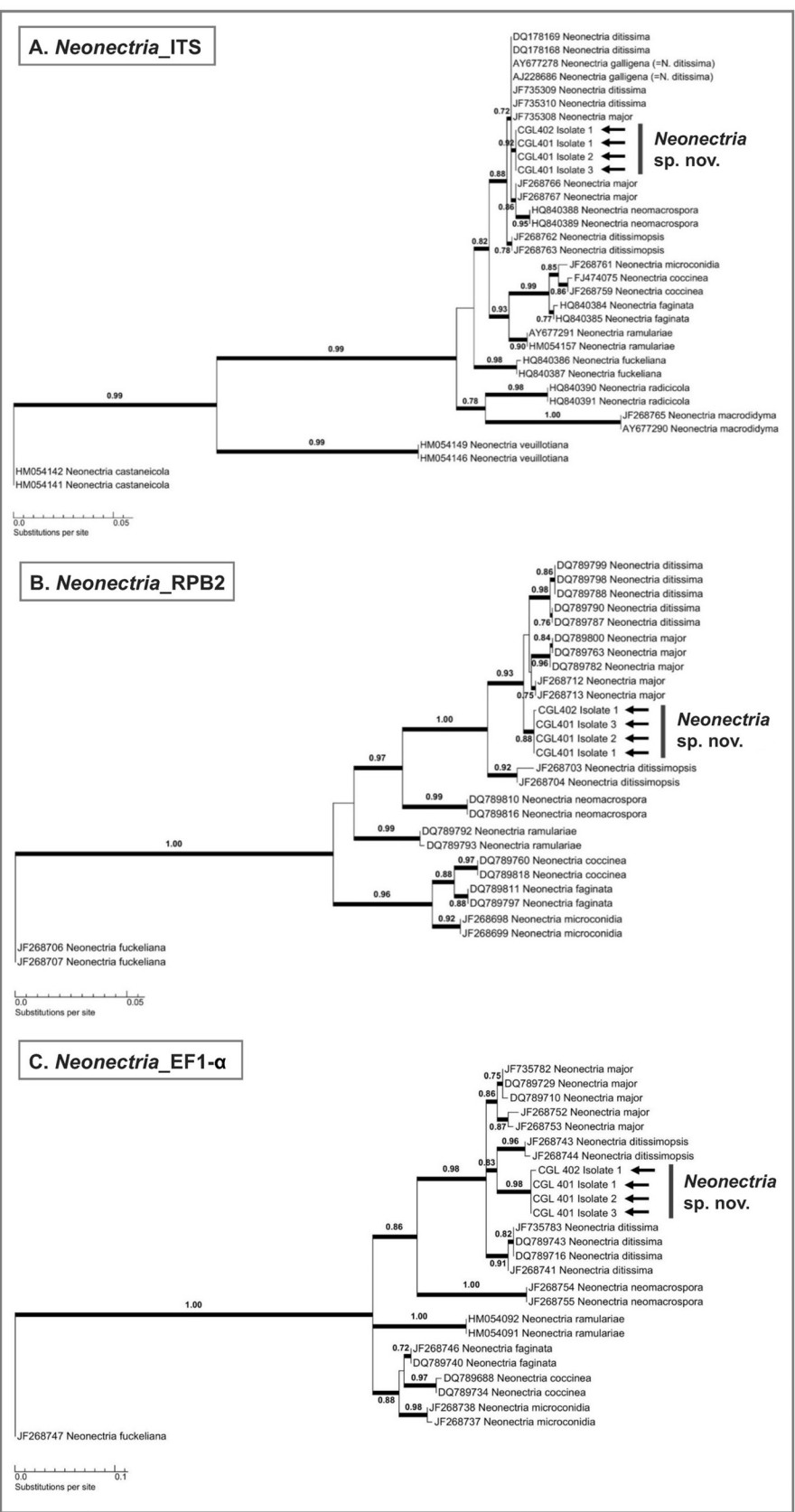

**Fig 3.** Single-locus maximum likelihood phylogenetic trees showing the phylogenetic placement of *Neonectria* sp. nov. isolates associated with *Alniphagus aspericollis* (marked by black arrows); (A) ITS region, (B) partial RPB2 region, and (C) partial EF1-α region. Only maximum likelihood values that exceed 0.65 are shown among the branches and are indicated by bold lines. GenBank accession numbers of previously published *Neonectria* spp. sequences are included.

The first group of *Ophiostoma* isolates (strains CGL101 to CGL113) was identified as *O. quercus* by phylogenetic analysis. The identity of this group could not be fully resolved to the species-level by analysis of the ITS barcode region, as it did not resolve separately from species from the "*O. ulmi* complex" (Fig 5A). Analysis of the partial BT gene region placed these isolates into two clades that grouped together with *O. quercus* isolates to form a single well-supported clade (Fig 5B). Similarly, analysis of the partial EF1-α gene region grouped all isolates with *O. quercus* to form a single well-supported clade (Fig 5C).

The second group of *Ophiostoma* isolates (strain CGL201) was identified by phylogenetic analysis as a putatively novel *Ophiostoma* sp. During analysis of the ITS barcode region, this group formed a single well-supported clade that was related to, but distinct from, *O. borealis*, *O. bacillisporum*, and *O. catonianum* (Fig 5A). Similarly, analysis of the partial BT gene region placed this group in a distinct and well-supported clade that was related to, but separate from, *O. catonianum* (Fig 5B). Phylogenetic analysis of the partial EF1-α gene region also placed this group in a single well-supported clade that was related to, but distinct from, *O. karelicum* (Fig 5C). The consistent, well-supported distinction of this group from other species in the "*O. ulmi* complex" strongly suggests the identification of an *Ophiostoma* species that has yet to be characterized.

Isolates from the genus *Cadophora* (strains CGL301 to CGL303) were identified as *C. spadicis*, based on comparison of ITS, EF1-α and BT sequences with those of closely-related *Cadophora* species [50] (Fig 6). DNA amplification of *Cadophora* isolates resulted in fragments of ~500 bp for the ITS barcode region and ~500 bp for the partial EF1-α gene region. Phylogenetic analysis of the ITS barcode region grouped all isolates with *C. spadicis* to form a single well-supported clade (Fig 6A). Analyses of the partial EF1-α gene region and the partial BT gene region for *Cadophora* isolates produced similar results to align with *C. spadicis* (Fig 6B, 6C).

The sole isolate from the ophiostomatoid genus *Graphilbum* was identified as *G. fragrans* based upon the closest BLAST matches of its ITS barcode sequence (Table 1) and culture morphology (e.g., production of yellow pigmentation in MEA, as described by [65, 66]) (Fig 4E). Single isolates of *Valsalnicola oxystoma* and *Chondrostereum purpureum* were also identified to the species-level by BLAST searches of their ITS barcode sequences (Table 1). Given that *G. fragrans*, *V. oxystoma* and *C. purpureum* were each only represented by one isolate, and no comprehensive genera-level phylogenetic studies are currently available for these species, no further phylogenetic analyses were conducted for these isolates.

## Relative prevalence of filamentous fungus species

*Neonectria* sp. nov., *O. quercus*, *Ophiostoma* sp. nov., *C. spadicis*, and *Penicillium* spp. were each isolated from both adult beetles and gallery phloem samples (Table 2). Additionally, *Graphilbum fragrans*, *Beauveria* spp., *Cladosporium* sp., *Acremonium* sp., and *Cosmospora* sp. were isolated only from adult beetles, while *Chondrostereum purpureum*, *Valsalnicola oxystoma*, *Pezicula* sp., and *Umbelopsis* sp. were isolated only from gallery phloem samples (Table 2).

*Neonectria* sp. nov. was the most frequently isolated filamentous fungus in our study, associated with approximately two-thirds of adult beetles and gallery phloem samples and almost all beetle-infested trees (Table 2). This canker fungus was concurrently isolated from both

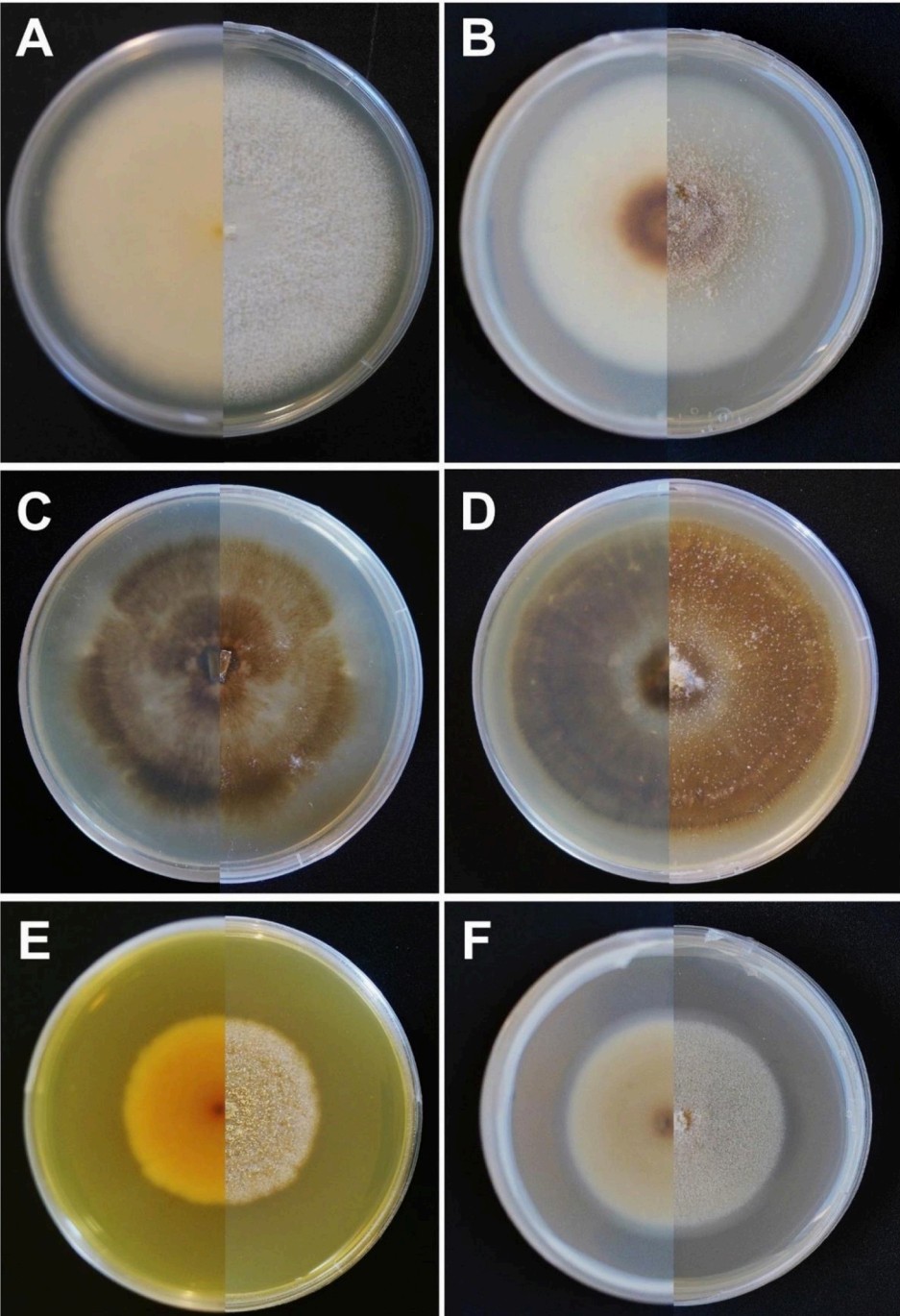

**Fig 4. Front and reverse sides of malt extract agar cultures of various filamentous fungi associated with
*Alniphagus aspericollis*.** (A) *Neonectria* sp. nov (*N. major*-like), (B,C) *Ophiostoma quercus*, (D) *Ophiostoma* sp. nov.,
(E) *Graphilbum fragrans*, and (F) *Cadophora spadicis*. Cultures were photographed following variable growth periods.

adult beetles and phloem samples collected from more than half of the sampled galleries. Aside
from Pacific Spirit Regional Park, and Tynehead Regional Park in the case of phloem isola-
tions, *Neonectria* sp. nov. was isolated from more than 50% of adult beetles and gallery phloem

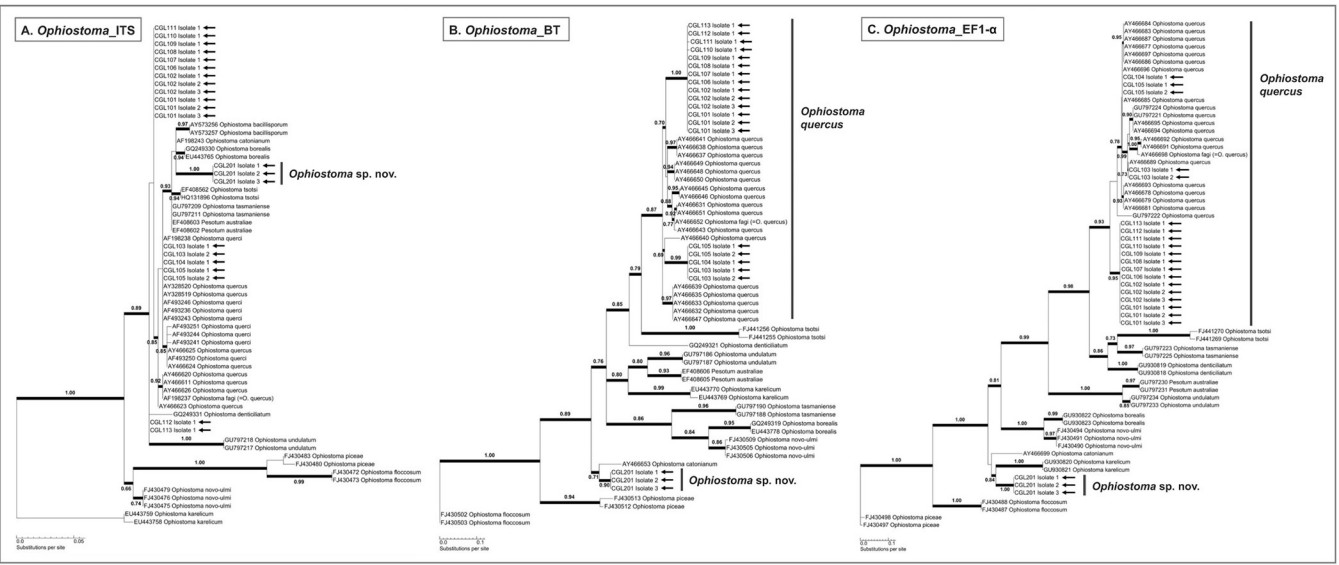

**Fig 5.** Single-locus maximum likelihood phylogenetic trees indicating the phylogenetic placement of *Ophiostoma* strains associated with *Alniphagus aspericollis*; (A) ITS region, (B) partial BT region, and (C) partial EF1-α region. Isolate sequences from this study, marked with black arrows, were compared to various *Ophiostoma* spp. in the "*O. ulmi* complex". Maximum likelihood values that exceed 0.65 are shown among the branches and are denoted by bold lines. GenBank accession numbers of previously published sequences are included.

samples collected at each study site (Table 3). Strikingly, *Neonectria* sp. nov. was isolated from every beetle-infested tree at each site except Pacific Spirit Regional Park, where two out of the three sampled trees yielded *Neonectria* sp. nov. isolates. At each site, however, isolation of *Neonectria* sp. nov. never exceeded 90% for either adult beetle isolations or gallery phloem samples. *Neonectria* sp. nov. was only occasionally isolated from *A. aspericollis* larvae, pupae, and teneral adults.

*Ophiostoma quercus* was the second most frequently isolated filamentous fungus, but its association with *A. aspericollis* was much less consistent than *Neonectria* sp. nov. Additionally, *O. quercus* was concurrently isolated from adult beetles and phloem samples from only 5.6% of sampled galleries (Table 2), nearly ten times lower than that for *Neonectria* sp. nov. At five of seven study sites, *O. quercus* was consistently isolated from 33% of adult beetles, with the exceptions of Pacific Spirit Regional Park and Cypress Mountain where isolation rates were 56% and 11%, respectively (Table 3). Isolation frequency of *O. quercus* was even lower for gallery phloem samples from each site, with three sites yielding no isolates of *O. quercus* from phloem at all. Beetle-infested trees surveyed at five of seven sites produced no isolates of *O. quercus*, and at Aldergrove Regional Park, *O. quercus* was not isolated from any beetle or gallery phloem sample. *Ophiostoma quercus* was almost never isolated from larvae, pupae, or teneral adults, the only exception being a single larva-derived isolate from Tynehead Regional Park.

*Ophiostoma* sp. nov. was rarely isolated, with an overall occurrence of just 3.7–16.7% on adult beetles, in galleries, and infested trees (Table 2). This was primarily because *Ophiostoma* sp. nov. was only isolated from Burnaby Mountain, where 33% of adult beetles and 22% of gallery phloem samples yielded the fungus (Table 3). *Ophiostoma* sp. nov. was isolated from all 3 beetle-infested trees surveyed at Burnaby Mountain but not concurrently isolated from adult beetles and gallery phloem samples from the same gallery. This putatively novel species of *Ophiostoma* was never isolated from larvae, pupae, or teneral adults.

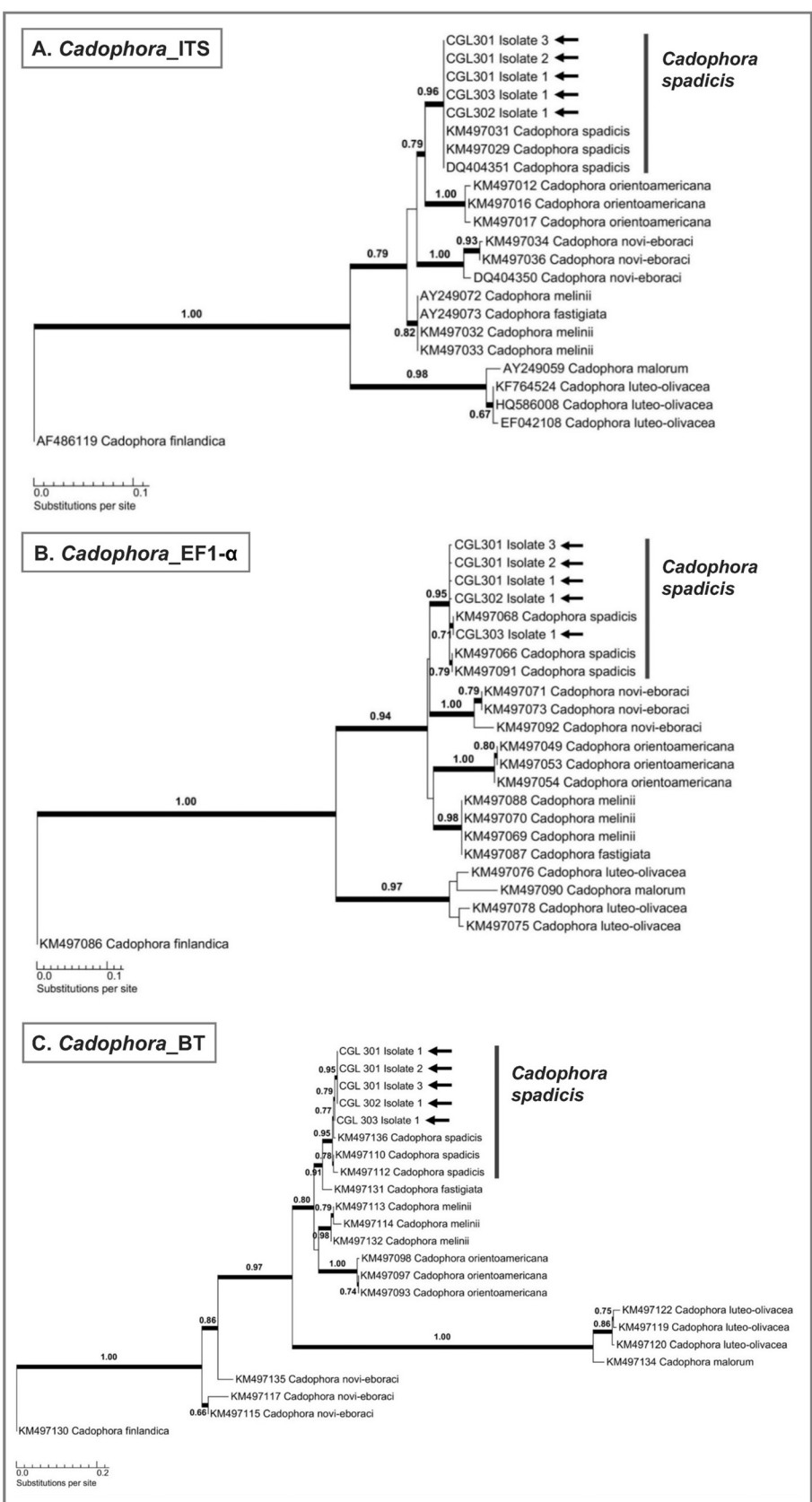

**Fig 6.** Single-locus maximum likelihood phylogenetic trees showing the phylogenetic placement of *Cadophora* isolates associated with *Alniphagus aspericollis* (marked by black arrows); (A) ITS region, (B) partial EF1-α region, and (C) partial BT region. Maximum likelihood values which exceed 0.65 are shown among the branches and are indicated by bold lines. GenBank accession numbers of previously published sequences are included.

*Cadophora spadicis* and *Penicillium* spp. were the two most prominent filamentous fungal taxa that did not fall into *Neonectria* or ophiostomatoid fungus categories. *Cadophora spadicis* was the third most frequently isolated filamentous fungus species, yet it was never concurrently isolated from beetles and gallery phloem samples from the same gallery (Table 2). *Penicillium* spp., comprising up to 10 different species in our study, had an overall occurrence of 7.4–38.9% (Tables 1 and 2). All other observed filamentous fungus species were rarely isolated. Except *Beauveria* spp., which occurred on 5.6% of adult beetles, these fungi were each isolated from only one or two adult beetle or gallery phloem samples (Table 2).

## Discussion

We found the alder bark beetle, *A. aspericollis*, to be associated with a diverse assemblage of filamentous fungus species that differed markedly in composition when compared with conifer tree-killing bark beetles [5, 20, 29]. The most frequently isolated fungus from *A. aspericollis*, gallery phloem, and nearly all beetle-infested trees was *Neonectria* sp. nov., a heretofore unknown type of ascomycete bark beetle associate that is most closely related to *N. major* (Wollenw.) Castl. & Rossman (Hypocreales: Nectriaceae). The fungal genus *Neonectria* includes many species that are well-known canker pathogens of trees; *N. major* is specific to

**Table 3. Prevalence of filamentous fungi associated with *Alniphagus aspericollis* including *Neonectria* sp. nov., *Ophiostoma quercus*, and *Ophiostoma* sp. nov. at each of seven sampling sites throughout the Greater Vancouver region (British Columbia, Canada).**

| Fungus sp. (by site) | Adult beetles | Gallery phloem | Concurrent isolations | Trees | Larvae | Pupae | Teneral adults |
|---|---|---|---|---|---|---|---|
| ***Neonectria* sp. nov.** | | | | | | | |
| Cypress Mountain | 7 (9) | 5 (9) | 5 (9) | 3 (3) | - | - | 0 (1) |
| Pacific Spirit Regional Park | 3 (9) | 2 (9) | 2 (9) | 2 (3) | 0 (1) | - | - |
| Tynehead Regional Park | 5 (9) | 4 (9) | 3 (9) | 3 (3) | 0 (2) | 1 (1) | - |
| Aldergrove Regional Park | 7 (9) | 6 (9) | 6 (9) | 3 (3) | 1 (1) | 2 (3) | 0 (1) |
| Gates Park | 5 (6) | 4 (6) | 4 (6) | 2 (2) | 0 (2) | - | 1 (1) |
| Burnaby Lake Regional Park | 2 (3) | 3 (3) | 2 (3) | 1 (1) | - | - | - |
| Burnaby Mountain | 7 (9) | 8 (9) | 7 (9) | 3 (3) | 1 (5) | - | - |
| ***Ophiostoma quercus*** | | | | | | | |
| Cypress Mountain | 1 (9) | 1 (9) | 1 (9) | 1 (3) | - | - | 0 (1) |
| Pacific Spirit Regional Park | 5 (9) | 1 (9) | 1 (9) | 2 (3) | 0 (1) | - | - |
| Tynehead Regional Park | 3 (9) | 2 (9) | 1 (9) | 2 (3) | 1 (2) | 0 (1) | - |
| Aldergrove Regional Park | 0 (9) | 0 (9) | 0 (9) | 0 (3) | 0 (1) | 0 (3) | 0 (1) |
| Gates Park | 2 (6) | 1 (6) | 0 (6) | 2 (2) | 0 (2) | - | 0 (1) |
| Burnaby Lake Regional Park | 1 (3) | 0 (3) | 0 (3) | 1 (1) | - | - | - |
| Burnaby Mountain | 3 (9) | 0 (9) | 0 (9) | 2 (3) | 0 (5) | - | - |
| ***Ophiostoma* sp. nov.** | | | | | | | |
| Burnaby Mountain | 3 (9) | 2 (9) | 0 (9) | 3 (3) | 0 (5) | - | - |
| Other sites | 0 (45) | 0 (45) | 0 (45) | 0 (15) | 0 (6) | 0 (4) | 0 (3) |

Frequency data are shown for isolations from adult beetles and gallery phloem samples, galleries with concurrent isolations from both adult beetles and phloem samples, beetle-infested trees with isolations from either adult beetles or phloem samples, and isolations from larvae, pupae and teneral adults. For each column category, values in parentheses refer to the total sample size per site.

the host genus *Alnus* [47] and forms perennial target-shaped stem cankers [40, 67] that may lead to limb damage and dieback over several years. *Neonectria major* has previously been reported from red alder stands in the Pacific Northwest, as well as from *Alnus* spp. in Norway and France [40, 45, 47, 67], and has even been utilized as a biological control agent for red alder within conifer plantations in British Columbia [47, 68, 69]. *Neonectria* fungi do not appear to possess sticky conidia or ascospores such as those of ophiostomatoid entomochoric fungi, and like *Geosmithia* spp. that are consistent associates of bark beetles, may not be specifically adapted for insect dispersal or employ alternate mechanisms for spore adhesion to the exoskeleton [28, 47, 70, 71].

*Ophiostoma quercus* (Georgev.) Nannf. (Ophiostomatales: Ophiostomataceae) was the second most frequently isolated fungal associate of *A. aspericollis*, yet it was infrequently isolated from adult beetles and even more rarely cultured from gallery phloem. *Ophiostoma quercus* is a sapwood-staining ophiostomatoid fungus that occurs worldwide in a variety of hardwood genera (e.g., *Quercus*, *Fagus*, *Acer*, *Betula*, *Eucalyptus*, *Castanea*) [60] and is considered to be saprotrophic or mildly pathogenic [72, 73]. *Ophiostoma quercus* has been found on *Alnus incana* in Latvia [74], but our study is the first to report *O. quercus* from *A. rubra*. As a casual associate of many hardwood-invading bark beetles worldwide [75, 76], it was probable that *O. quercus* would be occasionally isolated from *A. aspericollis*. Our study also reports the discovery of a new ophiostomatoid species, *Ophiostoma* sp. nov., a minor associate of *A. aspericollis*. *Ophiostoma* sp. nov. was only found at one of seven sampling sites, Burnaby Mountain, where it was isolated slightly more frequently than *O. quercus* from all beetle-infested trees but only occasionally cultured from adult *A. aspericollis* and gallery phloem. Many bark beetle species have more than one ophiostomatoid fungal partner [76–78], and the relative dominance of different ophiostomatoid partners may vary with geographic location and environmental conditions [28, 79].

Other tree pathogens and decay fungi, including *V. oxystoma* and *C. spadicis*, as well as various ubiquitous saprotrophs and the entomopathogenic *Beauveria* spp., were identified on *A. aspericollis* and in gallery phloem. These fungi were rarely isolated, however, and were obtained from either adult beetles or gallery phloem samples, but not both. For instance, *Beauveria* spp. were isolated from adult beetles only, which is consistent with the entomopathogenic function of these fungi [80, 81]. Likewise, fungi such as *V. oxystoma*, for which *A. rubra* is a known host, and *C. purpureum* were only isolated from gallery phloem samples, which is consistent with their ecological roles as tree pathogens and decay fungi [82, 83].

This study represents the first record of *C. spadicis* (Prodi, Sandalo, Tonti, Nipoti & A. Pisi) Travadon, Lawrence, Rooney-Latham, Gubler, Wilcox, Rolshausen & K. Baumgartner (Order: Helotiales) on red alder, and to the best of our knowledge, in western Canada. The genus *Cadophora* comprises various plant pathogens, decay fungi, and saprotrophs [50, 53, 84, 85]. *Cadophora spadicis* has been reported from grapevine, *Vitis* spp., in North America [50], is known to induce trunk hypertrophy in kiwifruit, *Actinia deliciosa*, in Italy [50, 53], and is associated with hardwood-infesting longhorn beetles (Coleoptera: Cerambycidae) in Finland [85]. Other *Cadophora* species have been occasionally isolated from *Alnus incana* and *A. glutinosa* [74, 83]. *Cadophora spadicis* was infrequently isolated from *A. aspericollis* and gallery phloem samples and could be a decay or pathogenic fungus of red alder that occasionally utilizes the entrance holes of *A. aspericollis* for access to host trees, and may therefore be coincidentally carried by the beetles. This passive association is also the likely case for other generalist saprotrophs (e.g., *Penicillium* spp., *Cladosporium* sp.) that were isolated during this study. These ubiquitous wind-dispersed fungi have been found in similar frequencies in other bark beetle systems [77, 86] and are not known to colonize living trees nor to be antagonists of bark beetles, unlike those reported by [87, 88] (e.g., *Trichoderma harzianum*, *T. polysporum*, *Aspergillus*

*fumigatus*, *A*. *nomius*). The ubiquitous saprophytic species identified by this study are therefore likely to be late colonizers of *A*. *aspericollis*-killed trees, or opportunistic hitchhikers of *A*. *aspericollis*, and are probably of little ecological importance to the beetle.

In keeping with our initial prediction, *A*. *aspericollis* did not display a consistent relationship with ophiostomatoid fungi but was instead casually associated with several species (i.e., *O*. *quercus*, *Ophiostoma* sp. nov., and *G*. *fragrans*). Conifer-killing bark beetles tend to be consistently associated with specific ophiostomatoid fungi [20, 29], with the possible exception of a few resource-pulse driven species (e.g., *Tomicus piniperda*, *Hylurgus ligniperda*) [77, 89]. Given that the host tree of *A*. *aspericollis*, *A*. *rubra*, fixes atmospheric nitrogen and its tissues are thereby rich in this limiting nutrient [37], selection for an association with ophiostomatoid fungi based upon their capacity to concentrate nitrogen in the phloem of their host trees [22, 23] seems unlikely. The absence of an intimate association between *A*. *aspericollis* and ophiostomatoid fungi lends support to the assertion that the mutualistic relationship between conifer-killing bark beetles and ophiostomatoid fungi arose in part from the fungi's capacity to improve the nutritional condition of phloem for developing beetles [21–23]. The loose association between *A*. *aspericollis* and ophiostomatoid fungi strongly suggests that these fungi do not provide any significant ecological benefits to *A*. *aspericollis*.

*Neonectria* sp. nov. was frequently isolated from both adult beetles and phloem within the same gallery, suggestive of a fungus–vector relationship where *A*. *aspericollis* is capable of transmitting *Neonectria* sp. nov. between host red alder trees. It is unclear, however, if *Neonectria* sp. nov. has an obligate or facultative relationship with *A*. *aspericollis* since the fungus was not isolated at the high frequencies (i.e., 90–100%) observed in well-established bark beetle–fungus symbioses (e.g., *Dendroctonus rufipennis* and *Leptographium abietinum*, *Dendroctonus ponderosae* and *Ophiostoma* spp., *Scolytus scolytus* and *O*. *ulmi*) [90–92]. Little is known about the etiology of *N*. *major*, but reproductive spores of the closely-related *Neonectria ditissima* (Tul. & C. Tul.) Samuels and Rossman are usually wind or water-dispersed, gaining entry to host trees via stem wounds, bark cracks, and leaf scars [93]. Cootsona [67] suggested that drought stress predisposes red alder to *N*. *major* infection in the Pacific Northwest and did not discount the possibility that insects could vector *N*. *major*. The pathogenicity of *Neonectria* sp. nov. remains untested, and we observed during sample collection that *Neonectria* sp. nov. infection in *A*. *aspericollis*-infested red alder was not accompanied by typical *Neonectria* disease symptoms. Canker pathogens including *Neonectria* and *Geosmithia* species usually induce target-shaped cankers or prominent lesions to form on the limbs or stems as infections progress [67, 93, 94]. None of the red alders sampled during this study displayed prominent canker or lesion symptoms on their main stems, raising the possibility that *Neonectria* sp. nov. may proliferate throughout tree tissues in a different manner when transmitted by *A*. *aspericollis* than when the fungus infects a host on its own.

Hardwood-infesting bark beetles maintain relationships of varying degrees of intimacy with both ophiostomatoid fungal associates and non-ophiostomatoid fungi including representatives from the order Hypocreales, the taxon that includes *Neonectria* and *Geosmithia*. *Scolytus scolytus* nearly always carries *O*. *ulmi* as part of the Dutch elm disease complex [90], while *S*. *multistriatus* and *S*. *schevyrewi* are associated with *O*. *novo-ulmi* at inconsistent frequencies (i.e., 9–92%) [95]. The birch bark beetle *S*. *ratzeburgi* is consistently associated with *O*. *karelicum*, a potentially pathogenic fungus that may be etiologically similar to Dutch elm disease [59, 96]. *Scolytus intricatus* and *Hylesinus varius*, which kill their respective weakened oak and ash hosts, are loosely and infrequently associated with ophiostomatoid fungi, including the non-virulent *O*. *quercus* [75, 97]. *Pseudopityophthorus* spp. are known to transmit oak wilt, *B*. *fagacearum* [32, 98]. The walnut twig beetle, *Pityophthorus juglandis*, vectors the phytopathogen *Geosmithia morbida*, a beetle–fungus complex responsible for the thousand canker disease

of black walnut in the western United States, yet in most cases *Geosmithia* spp. associates of bark beetles are not considered to be pathogenic [71, 94]. *Ceratocystis* spp. thought to be vectored by the mango bark beetle, *Hypocryphalus mangiferae*, cause diseases of mango trees in plantations worldwide [99–101].

Future research should focus on clarifying the ecological relationship between *A. aspericollis* and its *Neonectria* sp. nov. partner, specifically whether this association is strictly casual or representative of a symbiosis that is mutualistic, commensalistic, or antagonistic in nature. To determine whether *A. aspericollis* is a vector of *Neonectria* sp. nov., experimental work should be directed toward addressing the criteria outlined by Leach [102] for defining a pathogen–vector relationship. Further, assessing the effects of *Neonectria* sp. nov. on the host colonization and reproductive success of *A. aspericollis* would reveal whether this potential symbiosis represents a mutualism. The *Neonectria* sp. nov. isolates obtained during this study were nearly genetically identical, suggesting that this species originated from a single and recent evolutionary event that is worthy of future investigation. Finally, additional work is required to morphologically describe the novel species of *Neonectria* and *Ophiostoma* identified by phylogenetics herein.

## Supporting information

**S1 Table. Strains of *Neonectria* sp. nov., *Ophiostoma quercus*, *Ophiostoma* sp. nov., and *Cadophora spadicis* isolated from *Alniphagus aspericollis* and gallery phloem, including their collection site(s) and GenBank accession numbers for barcode sequences (internal transcribed spacer (ITS) rDNA gene region, partial beta-tubulin (BT) gene, partial translation elongation factor 1-alpha (EF1-α) gene, and partial RNA polymerase II second largest subunit (RPB2) gene).** Mycelia from each isolate comprising the listed strains, with the exception of several *O. quercus* isolates that were not successfully sub-cultured after initial investigation, were preserved in 10% glycerol, flash-frozen in liquid nitrogen, and placed at -80˚C for long-term storage in the Genomics and Forest Health (Hamelin) Lab at the University of British Columbia, Canada.
(DOCX)

## Acknowledgments

The authors thank members of the Forest Pathology Lab (Department of Forest and Conservation Sciences, UBC) including Hesther Yueh, Nicolas Feau, Dario Isidrio Ojeda, Padmini Herath, Sandra Cervantes, Monique Sakalidis, Angela Dale, Charlie Zha, Winnie Ho, and Barbara Wong, for their invaluable guidance; members of the Mycology Lab (Department of Botany, UBC) including Dr. Mary Berbee, Ludovic Le Renard, and Anna Bazzicalupo, for permission to use their equipment on occasion; and the Metro Vancouver Regional Parks Office, for permission to conduct field sampling.

## Author Contributions

**Conceptualization:** Gervais Y. S. Lee, Allan L. Carroll, Richard C. Hamelin.

**Data curation:** Gervais Y. S. Lee, Debra L. Wertman.

**Formal analysis:** Gervais Y. S. Lee, Debra L. Wertman, Richard C. Hamelin.

**Funding acquisition:** Allan L. Carroll, Richard C. Hamelin.

**Investigation:** Gervais Y. S. Lee, Allan L. Carroll, Richard C. Hamelin.

**Methodology:** Gervais Y. S. Lee, Debra L. Wertman, Allan L. Carroll, Richard C. Hamelin.

**Project administration:** Allan L. Carroll, Richard C. Hamelin.

**Resources:** Allan L. Carroll, Richard C. Hamelin.

**Software:** Richard C. Hamelin.

**Supervision:** Allan L. Carroll, Richard C. Hamelin.

**Validation:** Debra L. Wertman, Allan L. Carroll.

**Visualization:** Gervais Y. S. Lee, Allan L. Carroll, Richard C. Hamelin.

**Writing – original draft:** Gervais Y. S. Lee, Debra L. Wertman, Allan L. Carroll, Richard C. Hamelin.

**Writing – review & editing:** Debra L. Wertman, Allan L. Carroll, Richard C. Hamelin.

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
