## [Decision Letter · Decision Letter 0]

27 Dec 2022

PONE-D-22-28596A novel type of bark beetle–fungus association between the hardwood-killing alder bark beetle, Alniphagus aspericollis, and a Neonectria canker pathogenPLOS ONE

Dear Dr. Carroll,

Thank you for submitting your manuscript to PLOS ONE. After careful consideration, we feel that it has merit but does not fully meet PLOS ONE’s publication criteria as it currently stands. Therefore, we invite you to submit a revised version of the manuscript that addresses the points raised during the review process.

We look forward to receiving your revised manuscript.

Kind regards,

Simon Francis Shamoun, Ph.D.

Academic Editor

PLOS ONE

Journal Requirements:

“This research was funded by Natural Sciences and Engineering Research Council of Canada Discovery Grants to ALC (RGPIN-2015-04376) and RCH (RGPIN-2015-05279).”

“ALC: RGPIN-2015-04376, Natural Sciences and Engineering Research Council of Canada, Discovery Grant, https://www.nserc-crsng.gc.ca/

RCH: RGPIN-2015-05279, Natural Sciences and Engineering Research Council of Canada, Discovery Grant, https://www.nserc-crsng.gc.ca/

The funding agency did not play any role in the study design, data collection and analysis, decision to publish, or preparation of the manuscript.”

5. We note that Figure 2 in your submission contain map images which may be copyrighted. All PLOS content is published under the Creative Commons Attribution License (CC BY 4.0), which means that the manuscript, images, and Supporting Information files will be freely available online, and any third party is permitted to access, download, copy, distribute, and use these materials in any way, even commercially, with proper attribution. For these reasons, we cannot publish previously copyrighted maps or satellite images created using proprietary data, such as Google software (Google Maps, Street View, and Earth). For more information, see our copyright guidelines: http://journals.plos.org/plosone/s/licenses-and-copyright.

      1. You may seek permission from the original copyright holder of Figure 2 to publish the content specifically under the CC BY 4.0 license. 

Reviewers' comments:

Reviewer's Responses to Questions

**Comments to the Author**

1. Is the manuscript technically sound, and do the data support the conclusions?

Reviewer #1: Yes

Reviewer #2: Yes

2. Has the statistical analysis been performed appropriately and rigorously? 

Reviewer #1: Yes

Reviewer #2: Yes

3. Have the authors made all data underlying the findings in their manuscript fully available?

Reviewer #1: Yes

Reviewer #2: Yes

4. Is the manuscript presented in an intelligible fashion and written in standard English?

Reviewer #1: Yes

Reviewer #2: Yes

5. Review Comments to the Author

Reviewer #1: The paper deals with the previously unknown association between a fungus and a bark beetle.

I read this paper with much interest and found it well written and generally satisfying. I think it deals with topics that, now, are of great interest to both entomologists and forest pathologists and therefore worth publishing.

However, I also found some minor inaccuracies that need to be resolved for a better overall result, and I also feel that the analysis of the literature pertaining to insect-fungal relationships should be expanded a bit so as not to come to hasty and inaccurate conclusions.

In the title I would talk about fungi associated with Alniphagus aspericollis instead of Neonectria alone since the work done deals with several fungal species.

I would also not call the species of Neonectria that authors identified here a "canker pathogen" because it is not proven to be one: Koch's postulates have not been fulfilled yet. Belonging to a pathogen genus do not automatically mean that it is a pathogen. This is also the case of other fungi belonging to genera that usually contain pathogens, but they are not. Ophiostoma quercus, for example, share the same genus of some of the most infamous pathogens, but it is not a pathogen.

In addition, the association the authors describe is not a “novel type of association”, but it is a “newly discovered” association.

I would also avoid specifying in the title that the alder bark beetle is a hardwood-killing, because in many cases they kill plant already stressed by other factors that probably would have died anyway.

So I suggest to change from “A novel type of bark beetle–fungus association between the hardwood-killing alder bark beetle, Alniphagus aspericollis, and a Neonectria canker pathogen” into: “A bark beetle-fungus association between the alder bark beetle, Alniphagus aspericollis, and a new species of Neonectria” or ”Fungi associated to the alder bark beetle Alniphagus aspericollis”.

These general comments are valid throughout the manuscript.

Regarding the Discussion chapter I would recommend to include not just cases of pathogens associated to insects, but also other guilds of fungi. In addition, if Geosmithia morbida is considered the agent of Thousand cankers disease, G. pallida is reported just in one case of being a pathogen (and not confirmed elsewhere), while in many other reports the ecological niche occupied by this fungus is not clear, but surely not a plant pathogen. See also the suggested literature. I think that the Discussion chapter should be reviewed in the light of an improved literature review.

Reviewer #2: The manuscript titled “A novel type of bark beetle–fungus association between the hardwood-killing alder bark beetle, Alniphagus aspericollis, and a Neonectria canker pathogen” describes ID and occurrence of fungi associated with red alder and a bark beetle in the Vancouver, Canada area. The paper is clear, well written, thorough and the results support the conclusions. While an analysis on community composition differences among beetles and phloem could have informed the discussion (e.g., contingency table chi-square and/or NMDS), I feel that the descriptive statistics and interpretation were clear.

Specific comments:

Ln 95: Check journal style guide to check whether there should be a space between numbers and units. Here and throughout MS.

Ln 97: What temperature were samples stored at?

Ln 186: What was the total number of fungi sent for identification and what was the number breakdown by tissue sampled (i.e., beetle wash, phloem chips)

Ln 433 (Fig. 4): Consider referring to all the panels in this figure in text – even if it is just in the Table 1.

6. PLOS authors have the option to publish the peer review history of their article (what does this mean?). If published, this will include your full peer review and any attached files.

Reviewer #1: No

Reviewer #2: No

---

## [Author Response · Author response to Decision Letter 0]

22 Mar 2023

Reviewer #1 comments

The paper deals with the previously unknown association between a fungus and a bark beetle. 

I read this paper with much interest and found it well written and generally satisfying. I think it deals with topics that, now, are of great interest to both entomologists and forest pathologists and therefore worth publishing.

However, I also found some minor inaccuracies that need to be resolved for a better overall result, and I also feel that the analysis of the literature pertaining to insect-fungal relationships should be expanded a bit so as not to come to hasty and inaccurate conclusions.

These general comments are valid throughout the manuscript:

1. In the title I would talk about fungi associated with Alniphagus aspericollis instead of Neonectria alone since the work done deals with several fungal species.

Response: We have updated the title of our manuscript to address this comment and additional issues raised below. The new title is: “Filamentous fungal associates of the alder bark beetle, Alniphagus aspericollis, including an undescribed species of Neonectria,” and the short title: “Fungal associates of the alder bark beetle include a species of Neonectria.”

2. I would also not call the species of Neonectria that authors identified here a "canker pathogen" because it is not proven to be one: Koch's postulates have not been fulfilled yet. Belonging to a pathogen genus do not automatically mean that it is a pathogen. This is also the case of other fungi belonging to genera that usually contain pathogens, but they are not. Ophiostoma quercus, for example, share the same genus of some of the most infamous pathogens, but it is not a pathogen.

Response: This is a very important point raised by Reviewer #1 and we have updated the title, abstract, and main text of the manuscript to resolve the issue of Neonectria pathogenicity. It will be important for us to address Koch’s postulates in our forthcoming work on the alder bark beetle–Neonectria system as well, and this article will serve as a solid foundation for these studies. We concur that fungal pathogenicity cannot be assumed based on phylogenetic relationships and have revised our manuscript throughout to more accurately reflect what is known regarding the lifestyles of different taxa.

3. In addition, the association the authors describe is not a “novel type of association”, but it is a “newly discovered” association.

Response: We agree with Reviewer #1 and have removed this phrasing from the title. See L547-549 of the Discussion Section for a more appropriate description of the alder bark beetle–Neonectria association.

4. I would also avoid specifying in the title that the alder bark beetle is a hardwood-killing, because in many cases they kill plant already stressed by other factors that probably would have died anyway.

Response: While we have removed “hardwood-killing” from the title, we respectfully disagree with Reviewer #1 on this point as the alder bark beetle satisfies the definition of a pulse-driven irruptive bark beetle (Howe et al. 2022, Oecologia). The beetle attacks and eventually kills standing, live trees that may be healthy or stressed, and we have clarified this point on L89-90.

5. So I suggest to change from “A novel type of bark beetle–fungus association between the hardwood-killing alder bark beetle, Alniphagus aspericollis, and a Neonectria canker pathogen” into: “A bark beetle-fungus association between the alder bark beetle, Alniphagus aspericollis, and a new species of Neonectria” or “Fungi associated to the alder bark beetle Alniphagus aspericollis”.

Response: We have updated the title accordingly, please see our response to Reviewer #1 point 1. 

Specific comments:

6. Regarding the Discussion chapter I would recommend to include not just cases of pathogens associated to insects, but also other guilds of fungi. In addition, if Geosmithia morbida is considered the agent of Thousand cankers disease, G. pallida is reported just in one case of being a pathogen (and not confirmed elsewhere), while in many other reports the ecological niche occupied by this fungus is not clear, but surely not a plant pathogen. See also the suggested literature. I think that the Discussion chapter should be reviewed in the light of an improved literature review.

Response: Our Discussion Section does review non-phytopathogenic fungal associates of bark beetles including saprotrophic/weakly pathogenic Ophiostoma quercus (L430-433), entomopathogenic Beauveria spp. (L465-467), Chondrostereum purpureum (L467-469) and Cadophora spadicis (L474-481) as decay agents, and saprotrophs including Penicillium spp., Cladosporium sp. (L481-532). 

We have addressed the consistent associations between bark beetles and non-pathogenic Geosmithia spp., in contrast with G. morbida, on L423-427 and L608-611. Upon additional review of the literature, we do observe that G. pallida is likely not the causal agent of foamy bark canker and due to the ambiguity in this system we have removed this sentence and the corresponding article from our discussion – we thank Reviewer #1 for bringing this issue to our attention.

PDF-embedded comments:

7. Ln 397: on red alder on A. aspericollis

Response: We have left this statement as written because Cadophora spadicis was isolated from gallery phloem as well as from adult beetles.

8. Ln 459: It should be here mentioned that fungi of the genus Geosmithia are intimately associated to bark beetles and in most cases are not pathogens, as also reported in part of the cited literature see also Kolarik et al 2005 doi:10.1017/S0953756205003965; 2007 doi:10.1016/j.mycres.2007.06.010; and 2008 DOI: 10.1007/s00248-007-9251-0). In addition they are reported as ambrosia fungi (Kolarik and Kirkendall, 2010 doi:10.1016/j.funbio.2010.06.005) or mycoparasites (Pepori et al 2017 https://doi.org/10.1007/s00248-017-1062-3)

Response: Please see response to Reviewer #1 point 6 as we have updated the manuscript Discussion Section to address the non-phytopathogenic Geosmithia associates of bark beetles. We found the information needed for our updated statements on L423-427 and L608-611 in our existing citations (including Bettini et al. 2012 as suggested by Reviewer #2), but plan to incorporate more of the suggested literature on the lifestyles and habits of scolytine-associated Geosmithia fungi into upcoming manuscripts.

Reviewer #2 comments

The manuscript titled “A novel type of bark beetle–fungus association between the hardwood-killing alder bark beetle, Alniphagus aspericollis, and a Neonectria canker pathogen” describes ID and occurrence of fungi associated with red alder and a bark beetle in the Vancouver, Canada area. The paper is clear, well written, thorough and the results support the conclusions. 

1. While an analysis on community composition differences among beetles and phloem could have informed the discussion (e.g., contingency table chi-square and/or NMDS), I feel that the descriptive statistics and interpretation were clear.

Response: We appreciate this comment from Reviewer #2 and will consider community composition analyses for future studies.

Specific comments:

2. Ln 95: Check journal style guide to check whether there should be a space between numbers and units. Here and throughout MS.

Response: It does appear that PLOS ONE articles contain spaces between numbers and time/length/volume units so we have added these throughout the manuscript. We will defer to the journal for additional spacing requirements.

3. Ln 97: What temperature were samples stored at?

Response: We have added text to the Methods Section stating that the samples were stored in the refrigerator at 4C until they were processed.

4. Ln 186: What was the total number of fungi sent for identification and what was the number breakdown by tissue sampled (i.e., beetle wash, phloem chips)

Response: Text has been added to the Methods Section to indicate that a total of 54 adult beetles and 54 phloem samples were collected from across the seven sites, with sample size breakdown information provided in Fig 2 and its caption.

5. Ln 433 (Fig. 4): Consider referring to all the panels in this figure in text – even if it is just in the Table 1.

Response: Fig 4A and 4E are referenced in the main text of manuscript as they relate to the physical description of the cultures therein. Since we do not describe the morphology of cultures of the other species, we have opted not to add more in-text references to the Fig 4 panels but believe that it is still appropriate to provide these panels in the manuscript (e.g., as reference information for future studies).

PDF-embedded comments:

6. For a better readability I would suggest to use the form “Neonectria sp. nov.” instead of “N. sp. nov.”

Response: We agree and have updated N. sp. nov. to Neonectria sp. nov., as well as O. sp. nov. to Ophiostoma sp. nov., throughout the manuscript.

7. Ln 359: Indeed not really, it was shown that hydrophobins present on the conidia surface can mediate the attachment to hydrophobic surfaces, as the chitinous exoskeleton of the insect vectors and the conidia themselves (Wösten 2001; Temple and Horgen 2000). This is the case for entomopathogenic fungi, such as Beauveria bassiana (Zhang et al. 2011), and Geosmithia (Bettini et al., 2012) where the adhesion of conidia to the host’s surface is mediated by nonspecific hydrophobic and electrostatic interactions involving hydrophobin rodlet layers on the conidial cell wall.

Response: We thank Reviewer #2 for these insights which are extremely relevant to our interpretation of the relationship between the alder bark beetle and the Neonectria fungus. We have removed “Surprisingly” from L407 as suggested but note that our emphasis was directed to the identification of a non-ophiostomatoid associate, although we acknowledge that this statement did not account for the widespread associations between Geosmithia spp. and bark beetles which has since been addressed elsewhere in the manuscript (see responses to Reviewer #1 points 6 and 8). We have added one of the recommended papers (Bettini et al. 2012) to our statement about Geosmithia and Neonectria fungi lacking the sticky spores characteristic of ophiostomatoids and allude to potential alternate mechanisms for spore adhesion (L423-427). We will certainly address the potential role of hydrophobins in exoskeletal spore adhesion in a forthcoming paper that focusses on alder bark beetle vectoring of Neonectria sp. nov. and other insect vectors of Hypocreales fungi.

---

## [Editor Report · Decision Letter 1]

31 Mar 2023

Filamentous fungal associates of the alder bark beetle, Alniphagus aspericollis, including an undescribed species of Neonectria

PONE-D-22-28596R1

Dear Dr. Carroll,

We’re pleased to inform you that your manuscript has been judged scientifically suitable for publication and will be formally accepted for publication once it meets all outstanding technical requirements.

Kind regards,

Simon Francis Shamoun, Ph.D.

Academic Editor

PLOS ONE
---

## [Editor Report · Acceptance letter]

28 Apr 2023

PONE-D-22-28596R1 

Filamentous fungal associates of the alder bark beetle, *Alniphagus aspericollis,* including an undescribed species of *Neonectria*

Dear Dr. Carroll:

I'm pleased to inform you that your manuscript has been deemed suitable for publication in PLOS ONE. Congratulations! Your manuscript is now with our production department. 

Kind regards, 

on behalf of

Dr. Simon Francis Shamoun 

Academic Editor

PLOS ONE